Cultural differences in music features across Taiwanese, Japanese and American markets

http://orcid.org/0000-0002-0755-7173 Liew Kongmeng 1 2 liew.kongmeng@is.naist.jp
Uchida Yukiko 3
http://orcid.org/0000-0003-4599-0442 de Almeida Igor 4
1 Graduate School of Science and Technology, Nara Institute of Science and Technology , Ikoma, Nara Prefecture , Japan
2 Graduate School of Human and Environmental Studies, Kyoto University , Kyoto, Kyoto Prefecture , Japan
3 Kokoro Research Center, Kyoto University , Kyoto, Kyoto Prefecture , Japan
4 Institute of Liberal Arts, Otemon Gakuin University , Ibaraki, Osaka Prefecture , Japan
Jung Jason
Electronic publication date: 2021 Aug 3
Publication date: 2021
Volume: 7
Electronic Location ID: e642
Received 2021 Mar 3; Accepted 2021 Jun 24
Copyright: © 2021 Liew et al.
Copyright year: 2021
Copyright holder: Liew et al.
License: This is an open access article distributed under the terms of the Creative Commons Attribution License, which permits unrestricted use, distribution, reproduction and adaptation in any medium and for any purpose provided that it is properly attributed. For attribution, the original author(s), title, publication source (PeerJ Computer Science) and either DOI or URL of the article must be cited.
License URL: https://creativecommons.org/licenses/by/4.0/

Keywords: Music, Culture, Psychology, Spotify, Machine Learning

Funding: Japan Society for the Promotion of Science 19J14431 This work was supported by a Grant-in-Aid for Research Fellows of the Japan Society for the Promotion of Science (19J14431). The funders had no role in study design, data collection and analysis, decision to publish, or preparation of the manuscript.

==============================
Background

Preferences for music can be represented through music features. The widespread prevalence of music streaming has allowed for music feature information to be consolidated by service providers like Spotify. In this paper, we demonstrate that machine learning classification on cultural market membership (Taiwanese, Japanese, American) by music features reveals variations in popular music across these markets.

Methods

We present an exploratory analysis of 1.08 million songs centred on Taiwanese, Japanese and American markets. We use both multiclass classification models (Gradient Boosted Decision Trees (GBDT) and Multilayer Perceptron (MLP)), and binary classification models, and interpret their results using variable importance measures and Partial Dependence Plots. To ensure the reliability of our interpretations, we conducted a follow-up study comparing Top-50 playlists from Taiwan, Japan, and the US on identified variables of importance.

Results

The multiclass models achieved moderate classification accuracy (GBDT = 0.69, MLP = 0.66). Accuracy scores for binary classification models ranged between 0.71 to 0.81. Model interpretation revealed music features of greatest importance: Overall, popular music in Taiwan was characterised by high acousticness, American music was characterised by high speechiness, and Japanese music was characterised by high energy features. A follow-up study using Top-50 charts found similarly significant differences between cultures for these three features.

Conclusion

We demonstrate that machine learning can reveal both the magnitude of differences in music preference across Taiwanese, Japanese, and American markets, and where these preferences are different. While this paper is limited to Spotify data, it underscores the potential contribution of machine learning in exploratory approaches to research on cultural differences.

Introduction

With 219 million active listeners a month, and a presence in over 60 countries, Spotify is one of the largest music streaming service providers in the world (as of 2020, Schwind et al., 2019). To facilitate such a service, they maintain a database of music features for all songs in their service, that is made publicly accessible through the Spotify API (Application Programme Interface). This database contains a wealth of meta and music feature data, that researchers have been using to research human behaviour and engagement with music. For example, Park et al. (2019) analysed data from one million individuals across 51 countries and uncovered consistent patterns of music preferences across day-night cycles: relaxing music was more commonly played at night, and energetic music during the day. Pérez-Verdejo et al. (2020) found that popular hit songs in Mexico shared many similarities to global hit songs. Spotify’s music features have also been used to examine songs in clinical and therapeutic settings, with Howlin & Rooney (2020) finding that songs used in previous pain management research, if chosen by the patient, tended to have high energy, danceability, and lower instrumentalness features, than experimenter-chosen songs.

In this paper, we use Spotify features to examine how cultures differ in music preferences, through a bottom-up, data-driven analysis of music features across cultures. Here, we quantify music preference through music features, following past research (e.g., Fricke et al., 2019). Music plays a huge role in human society, be it in emotion regulation or for social displays of identity and social bonding (Groarke & Hogan, 2018; Dunbar, 2012). These are often embedded in the cultural norms and traditions of the listener. In other words, understanding how music differs across cultures may reflect corresponding cultural differences in the sociocultural context that shape our individual preferences towards certain types of music over others. Thus, understanding how music differs between these cultural markets, e.g., by examining their features, may then shed light on possible cultural differences, which can guide follow-up research by generating novel hypotheses, or in supporting various theories on cultural differences aside from music (see below).

To achieve this aim, we rely on machine learning classification of songs based on cultural markets (i.e., culture of origin), as interpretation of these models may reveal insight into how (and which) features differ between cultures. We utilise data seeded originally on Taiwanese, American and Japanese Top-50 lists. This arguably aligns more towards a sociolingual distinction of cultural membership, and not a country-based sampling commonly used in culture research. Our reasoning was that this more closely resembles the differentiated cultural markets that artists of a certain language operate in, that often transcend country boundaries. For example, popular Chinese music meant for a Chinese cultural market often subsumes artists from wider Chinese cultural origins (in countries and territories like Taiwan, Hong Kong, Singapore and Malaysia; Fung, 2013; Moskowitz, 2008). As such, in this paper, these were considered to belong to a ‘Chinese’ cultural market (including music from language subtypes and dialects). Japanese music was similarly treated as belonging to a ‘Japanese’ market, and music from Western (Anglo-European) cultural origins (e.g., the US, UK, Canada, and Australia) were considered as belonging to a ‘Western’ cultural market.

Past approaches towards music preference through Spotify have largely focused on curated lists of Top-50 or Top-200 popular songs (e.g., Pérez-Verdejo et al., 2020; Febirautami, Surjandari & Laoh, 2019). While such lists are often diluted by the inclusion of ‘global’ hit songs, they nevertheless provide a window to examine culturally based music preferences. Accordingly, we also conduct a follow-up study using these Top-50 lists from Taiwan, Japan, and the US to ensure the reliability of our interpretations (from the classification model).

Cultural differences in music preference

Typically, most of the research in cultural differences in the psychological literature has come from top-down, theoretical approaches. These have been instrumental in shaping the field, by increasing awareness of systematic ways by which people from cultures are different. One of the most instrumental differences is in the independence and interdependence of the self (Markus & Kitayama, 1991, 2010). Westerners generally tend to be independent, in that they prioritise the autonomy and uniqueness of internal (self) attributes. In contrast, East Asians generally tend to be interdependent, where their concept of self is intricately linked to close social relationships. This has been shown to have implications on music preferences through differences in desirability of emotions. For example, Westerners generally tend to view happiness as a positive and internal hedonic experience to be maximised where possible (Joshanloo & Weijers, 2014). As such, Western music preference tends towards high-arousal music, possibly in the search of strong ‘happiness’ experiences. East Asians, however, view happiness as a positive feeling associated with harmonious social relationships, in contrast to the hedonic, high-arousal definition in Western contexts. Consistently, East Asian preferences for music do not have this high-arousal component, and is more calm, subdued and relaxing (Tsai, 2007; Uchida & Kitayama, 2009; Park et al., 2019).

However, such theoretically based analyses may overrepresent Western cultures in research and literature. Consequently, cultural differences within similar, non-Westernised spheres are not well understood, due to the lack of pre-existing theory. For example, Chinese and Japanese cultures are often grouped together in cross-cultural research as a representation of East Asian collectivism, that functions as a comparative antithesis to Western findings (e.g., Heine & Hamamura, 2007). Yet, research has also uncovered differences between China and Japan that cannot be explained by these theories (Muthukrishna et al., 2020). As such, cultural differences within East Asia are not well understood in the psychological literature, and few theories exist to offer predictions on differences in music preference between these cultures.

Our solution was to examine music as cultural products from the bottom-up. Doing so would reduce the effect of experimenter bias in guiding theory formation and interpretation, when examining a wide database of music features. Cultural products are behavioural manifestations of culture that embody the shared values and collective aesthetics of a society (Morling & Lamoreaux, 2008; Lamoreaux & Morling, 2012; Smith et al., 2013). This implies that music consumption behaviour underscores culturally based attitudes, cognitions and emotions that afford preferences for certain congruent types of music. For example, in a cross-cultural comparison between Brazil and Japan, De Almeida & Uchida (2018) found that Brazilian song lyrics contained higher frequencies of positive emotion words and lower frequencies of neutral words than Japanese lyrics. This was consistent and reflective of their respective cultural emphases on emotion expressions (see Triandis et al., 1984; Uchida & Kitayama, 2009), and showed that comparing music ‘products’ elucidated differences between the collective shared values of different cultures. Past research on cultural products have relied on both popularity lists (charts; e.g., Askin & Mauskapf, 2017), and on artifacts produced by a culture (e.g., Tweets: Golder & Macy, 2011; newspaper articles: Bardi, Calogero & Mullen, 2008). We utilise both methods, and propose that examining cultural differences in ‘music’ products on a large-scale may provide potential insight into the sociocultural circumstances that give rise to these differences.

The present research

We adopted a data-driven, bottom-up approach to explore music preferences between cultures/industries through musical features for this study through machine learning. i.e., we first train a multiclass model to classify songs as belonging to (originating from) Chinese (Taiwanese), Japanese, or Western (American/English) markets. This is to establish the presence and magnitude of discernible cultural differences in music features. Next, we decompose the model by training three binary machine learning classifiers to classify songs as belonging to one culture or another. By applying model interpretation techniques on these models (such as Partial Dependence Plots (PDPs)), we aim to discover the specific difference in preferred musical features between Chinese (Taiwanese)-Japanese markets, Chinese (Taiwanese)-Western (American/English) markets and Japanese–Western (American/English) markets. We aimed to include as many songs as possible that were produced from these respective culture-based music industries to observe systematic trends and differences from as wide a range of musical styles and genres within these industries as possible. Finally, we examine the generalizability of these interpretations by conducting a follow-up study on Top-50 songs from Taiwan, Japan, and the US. If the identified features of difference present in songs produced by a cultural market were indeed representative of cultural differences in music preferences, we expect that these features should also show consistent differences for their respective Top-50 (popular) songs.

Materials & Methods

Overview

To explore differences between cultures, we used machine learning to classify a database of Chinese, Japanese, and English songs into their respective cultural (linguistic) markets. As this was a multiclass classification problem, we conducted the analysis twice using gradient boosted decision trees (GBDTs) and artificial neural networks (multi-layer perceptron, MLP) that are inherently capable of multiclass classification. This was also to examine the consistency in results between two differing methods of analysis, and strengthen the reliability of the analyses. To infer the features that accounted for cultural differences, we use model interpretation techniques, namely relative feature importance (RFI; Friedman, 2001), permutational feature importance (PFI; Fisher, Rudin & Dominici, 2018), and partial dependence plots (PDPs, Friedman, 2001), to examine and visualise the relationships between the feature and its influence on the probability of classification.

Data mining

We accessed the Spotify Application Programme Interface (API) through the ‘spotifyr’ wrapper (Thompson, Parry & Wolff, 2019) in R, to obtain song-level music feature information from Chinese, Japanese and English artists from the Spotify database. This was through a pseudo-snowball sampling method: we relied on Spotify’s recommendation systems (the ‘get_related_artists’ function) to recommend artists related to those in the official Spotify Top-50 chart playlists for Taiwan, Japan, and the US respectively and created a list of artists per country. We then used the same method to obtain another list of recommended artists to these respective ‘lists’, for up to six iterations, in order to obtain comparable sample sizes between these three markets. We also excluded all non-Chinese, non-Japanese, and non-English (language) artists from the respective list. This was through an examination of the associated genres for each artist, which often contained hints to their cultural origins (e.g., J-pop, J-rock, Mandopop). Artists that did not have listed genres were checked manually by the researchers. This resulted in a final N(artists) = 10,259 (Japanese = 2,587; English = 2,466; Chinese = 5,206). All song-level feature information for all artists were then obtained from the Spotify database. Duplicates (such as the same song being rereleased in compilation albums) were removed, for a total of N(songs) = 1,810,210 (Japanese = 646,440; Chinese = 360,101; English = 803,669). To ensure class balances, we randomly downsampled the Japanese and English samples to match the Chinese sample, resulting in a final N(songs) = 1,080,303.

Data handling and analysis

Except for ‘key’ and ‘time signature’, all Spotify features were inputted as features in the classification models. These were: ‘danceability’, ‘energy’, ‘loudness’, ‘speechiness’, ‘acousticness’, ‘instrumentalness’, ‘liveness’, ‘valence’, ‘tempo’, ‘duration’ (ms), and ‘mode’. A list of definitions for these features is available in Table 1. These were to classify songs according to their cultural membership (Chinese, English, or Japanese), as the outcome variable. The data was split into a training and testing set along a 3:1 ratio. Parameters for the GBDT model and weights for the MLP model were tuned through 5-fold cross validation on the training set. We also examined RFI scores for each model. For GBDT, this was a measure of the proportion that a feature was selected for stratification in each iterative tree, and for MLP, this was based on PFI, which measures the resultant error of a model when each feature is iteratively shuffled—the greater the error, the larger the influence a feature exerts on the outcome variable (Fisher, Rudin & Dominici, 2018; Molnar, 2019). We then simplified the classification problem by splitting it into 3 separate binary classifications: Japanese–Chinese, Japanese–English and Chinese–English. GBDTs and MLP models were conducted for these three comparisons, and in addition to RFI measures, we visualised the effect of each variable using PDPs. These show the averaged marginal effect of a feature on the outcome variable in a machine learning model, and is useful to glean an understanding of the nature of the relationship between these variables. PDPs were conducted through the ‘pdp’ package (Greenwell, 2017), and PFIs were conducted through the ‘iml’ package (Molnar, Bischl & Casalicchio, 2019). Machine learning was conducted through the ‘gbm’ package (Greenwell et al., 2019) for GBDTs, and the ‘nnet’ package (Venables & Ripley, 2002) for MLPs, via the ‘caret’ wrapper (Kuhn, 2019) in R (R Core Team, 2019). All R scripts used for data mining and analysis are available in our OSF repository (https://osf.io/d3cky).

Table 1 A list of song-level audio features obtained from Spotify for our analyses.

Audio feature	Description	
Duration	The duration of the music in milliseconds (ms).	
Mode	If the melody of a track is in a major or minor key.	
Acousticness	A confidence measure on whether a song is acoustic.	
Danceability	The suitability of a song for dancing. This is based on several musical features, such as tempo, rhythmic stability, regularity, and beat strength.	
Energy	A measure of the intensity and activity of a song as perceptually lound, fast, or noisy. This is based on several musical and spectral features, such as dynamic range, loudness, timbre, onset rate, and entropy.	
Instrumentalness	A confidence measure of whether a song contains no vocals.	
Liveness	A confidence measure on the presence of audiences in the recording.	
Loudness	The overall intensity of the song in decibels (dBFS).	
Speechiness	A confidence measure on the presence of spoken words in a song.	
Valence	An estimate of whether a song conveys positive or negative affect.	
Tempo	The estimated main tempo of a song.	
Note:

Audio features refer to the music features as listed on the Spotify API, followed by a brief description of each feature. More information on the features are available at: https://developer.spotify.com/documentation/web-api/reference/tracks/get-audio-features/.

Results

Descriptives

Table 2 reports the descriptive medians, lower/upper quantiles, and missing data for each feature per culture. The full list of artists, genres, and songs are available in our OSF repository (https://osf.io/d3cky). Additionally, we note that while our database of songs spans as early as the 1950s, most of the songs in our database were from the mid-2000s to 2020 (see Fig. 1).

Figure 1 Number of songs (in our data) by year for Japanese (JP), English (US) and Chinese (ZH) medium songs.

Table 2 Medians (L/U quantiles) and missing data for musical features (excluding mode), and release year.

Feature	Chinese	Japanese	Western	
Median (L/U)	Missing	Median (L/U)	Missing	Median (L/U)	Missing	
Danceability	0.56 (0.46/0.66)	3	0.56 (0.45/0.67)	4	0.66 (0.55/0.76)	1	
Energy	0.46 (0.34/0.63)	3	0.76 (0.51/0.90)	4	0.69 (0.54/0.82)	0	
Loudness	−8.9 (−11.1/−6.9)	3	−6.2 (−9.3/−4.3)	4	−6.8 (−8.9/−5.2)	0	
Speechiness	0.04 (0.03/0.05)	3	0.05 (0.04/0.09)	4	0.08 (0.04/0.24)	0	
Acousticness	0.54 (0.21/0.76)	3	0.11 (0.01/0.49)	4	0.09 (0.02/0.29)	1	
Instrumentalness	1.4E−6 (0.00/1.1E−4)	3	1.0E−4 (0.00/0.36)	4	0.00 (0.00/0.0006)	1	
Liveness	0.13 (0.10/0.21)	3	0.14 (0.10/0.29)	4	0.14 (0.10/0.29)	1	
Valence	0.37 (0.24/0.57)	3	0.52 (0.31, 0.71)	4	0.51 (0.33/0.69)	1	
Tempo	122.7 (100.0/138.1)	3	123.7 (99.0/144.0)	4	119.7 (96.6.136.2)	0	
Duration (ms)	240,213 (205,586/272,586)	3	236,840 (187,266/281,573)	4	215,640 (184,727/250,693)	0	
Release year	2011 (2005/2015)	148	2015 (2010/2018)	100	2016 (2012/2018)	0	

Multiclass classification (Chinese–Japanese–English)

For the GBDT model, the parameter tuning resulted in N(trees) = 150, interaction depth = 3, alongside default parameters of shrinkage = 0.1, and number of minimum observations per node = 10. The GBDT achieved a classification accuracy of 0.682, 95% CI [0.680–0.683], significantly above the no information rate (NIR) of 0.333, p < 0.0001. Aside from the input and output layers, we used a MLP model consisting of one hidden layer with five nodes. The MLP model achieved a slightly lower accuracy score of 0.660, 95% CI [0.659–0.662], but was still significantly above the NIR of 0.333, p < 0.0001. RFIs for both models are reported in Table 3.

Table 3 Comparison of feature importance between the GBDT and MLP multiclass models.

Feature	RFI (GBDT)	PFI (MLP)	
Speechiness	24.3	1.26	
Loudness	7.4	1.15	
Instrumentalness	15.6	1.19	
Acousticness	16.5	1.14	
Energy	17.5	1.12	
Mode name	0.1	1.00	
Duration	8.4	1.12	
Danceability	5.9	1.06	
Valence	1.8	1.03	
Tempo	2.2	1.01	
Liveness	0.2	1.00	

Binary classifications

We first unpack the Chinese–Japanese model: For the GBDT model, the parameter tuning resulted in N(trees) = 150, interaction depth = 3, and no changes were made to the other default parameters (as above). The GBDT achieved a classification accuracy of 0.784, 95% CI [0.783–0.786], AUC = 0.865, significantly above the no information rate (NIR) of 0.500, p < 0.0001. The MLP model achieved a comparable accuracy score of 0.766, 95% CI [0.764–0.768], AUC = 0.844, significantly above the NIR of 0.500, p < 0.0001. Next, the Chinese–English model: For the GBDT model, the parameter tuning resulted in N(trees) = 150, interaction depth = 3, and no changes were made to the other default parameters. The GBDT achieved a classification accuracy of 0.807, 95% CI [0.805–0.809], AUC = 0.885, significantly above the no information rate (NIR) of 0.500, p < 0.0001. The MLP model achieved a comparable accuracy score of 0.803, 95% CI [0.801–0.805], AUC = 0.880, significantly above the NIR of 0.500, p < 0.0001. Finally, the Japanese–English model: For the GBDT model, the parameter tuning resulted in N(trees) = 150, interaction depth = 3, and no changes were made to the other default parameters. The GBDT achieved a classification accuracy of 0.713, 95% CI [0.711–0.715], AUC = 0.797, significantly above the no information rate (NIR) of 0.500, p < 0.0001. The MLP model achieved a comparable accuracy score of 0.709, 95% CI [0.707–0.711], AUC = 0.791, significantly above the NIR of 0.500, p < 0.0001. All RFIs and PFIs are reported in Table 4. Additionally, the two most important features are visualised by PDPs in Fig. 2. A visual inspection of the PDPs suggests that English music is higher than both Japanese and Chinese music in speechiness, Chinese music is higher than both Japanese and English music in acousticness, and Japanese music is higher than English and Chinese music in energy. In comparing Japanese and Chinese music, we note that acousticness and energy were also present, but were identified only in the GBDT model. In contrast, the MLP model identified loudness and instrumentalness as higher in Japanese music than Chinese music.

Figure 2 PDPs of top two most important features in each model on the probability of classification.

The positive class is indicated at the top of the Y-axis. For example, in the Japanese–English GBDT model (top right) for the speechiness feature, the decreasing trend indicates that the higher the speechiness score, the lower the probability of classification (of a song) as being Japanese (i.e., higher probability of being English), in a fairly linear fashion.

Table 4 Comparison of feature importance measures for respective binomial classification models.

Feature	Chinese–Japanese	Chinese–English	Japanese–English	
RFI (GBDT)	PFI (MLP)	RFI (GBDT)	PFI (MLP)	RFI (GBDT)	PFI (MLP)	
Speechiness	4.8	1.05	46.6	1.67	39.6	1.22	
Loudness	14.1	1.46	3.0	1.19	4.7	1.04	
Instrumentalness	26.0	1.22	2.8	1.08	8.5	1.105^	
Acousticness	31.1	1.16	34.1	1.35	2.4	1.04	
Energy	32.9	1.07	4.7	1.09	16.0	1.114^	
Mode name	0.0	1.00	0.0	1.01	0.2	1.00	
Duration	9.3	1.08	4.3	1.07	12.9	1.02	
Danceability	1.2	1.01	1.6	1.06	10.4	1.110^	
Valence	0.7	1.06	1.2	1.02	3.3	1.003	
Tempo	1.4	1.01	1.5	1.02	1.9	1.00	
Liveness	0.0	1.00	0.2	1.00	0.1	1.00	
Note:

While scales between RFI and PFI are not equivalent, both measure model-specific feature importance relative to other features: the higher the score, the larger the importance within the model. Features with highest importance are in bold. PFIs were reported with two decimal places, but we used three decimal places for PFIs denoted by ‘^’. This was to identify the 2nd most important feature for the PDP.

Overall, this suggests that, unlike English–Japanese or English–Chinese comparisons which were markedly different on a few main features, the differences between Chinese and Japanese music were spread widely across the various features. Consequently, despite relying on different ‘important variables’, both the MLP and GBDT managed to achieve a comparably high classification accuracy, with the GBDT outperforming the MLP for all classification tasks (results from DeLong’s tests are available on our OSF repository).

Additional analyses

We also visualised the changes in features over time for speechiness, acousticness, energy, instrumentalness, and loudness, from 2000 to 2020 (Fig. 3). Feature information for songs before 2000 were excluded due to the markedly smaller sample. Other than instrumentalness, which showed a notable decrease over time in Japanese songs, the remaining four features showed stability over time. This suggests that the differences in preference highlighted by the RFIs, PFIs and PDPs could indicate long term cultural preferences for music.

Follow-up study

We obtained a second round of data (approximately one year later) from Top-50 lists for Japan, Taiwan and the USA. Focusing on the identified features of speechiness, acousticness, energy, instrumentalness, and loudness from the previous study, Kruskal-Wallis tests revealed a significant effect of energy on speechiness (χ2(2) = 30.5, p < 0.001), acousticness (χ2(2) = 24.5, p < 0.001), energy (χ2(2) = 21.0, p < 0.001), and loudness (χ2(2) = 33.5, p < 0.001), but no significant effect was observed for instrumentalness (χ2(2) = 1.4, p = 0.49). For speechiness, post-hoc Dwass-Steel-Critchlow-Fligner pairwise comparisons revealed that USA was significantly higher than Taiwan (W = 7.34, p < 0.001) and Japan (W = 5.91, p < 0.001), but no significant difference was observed between Taiwan and Japan (W = -1.48, p = 0.55). For acousticness, Taiwan was significantly higher than Japan (W = 5.97, p < 0.001) and the USA (W = 6.11, p < 0.001), but no significant difference was observed between the USA and Japan (W = 0.53, p = 0.93). For energy, Japan was significantly higher than Taiwan (W = 6.01, p < 0.001), and the USA (W = 4.35, p = 0.006), but no significant difference was observed between Taiwan and the USA (W = 2.88, p = 0.103). For loudness, Japan was significantly higher than Taiwan (W = 7.44, p < 0.001) and the USA (W = 6.36, p < 0.001), but no significant difference was observed between Taiwan and the USA (W = 2.20, p = 0.27). Finally, for instrumentalness, no significant difference was observed between Japan and Taiwan (W = 0.61, p = 0.90), Japan and the USA (W = 1.67, p = 0.48), or Taiwan and the USA (W = 1.01, p = 0.75).

In short, with the exception of instrumentalness, Top-50 playlists obtained one year later nevertheless demonstrate strong consistency with the earlier results. American Top-50 songs are higher than both Japanese and Taiwanese Top-50 songs in speechiness, Taiwanese Top-50 songs are higher than both Japanese and American Top-50 songs in acousticness, and Japanese Top-50 songs are higher than American and Taiwanese music in energy and loudness. However, instrumentalness, that was originally identified as a variable of importance for the MLP model, did not consistently differ between cultures. Indeed, Fig. 3 shows that instrumentalness in Chinese music is inconsistent, with strong fluctuations depending on year. More research is needed to determine if instrumentalness is indeed a preferred feature in Taiwanese markets or merely a passing trend.

Figure 3 Variation and stability of features (medians) across cultures from 2000 to 2020.

Note that scales differ bewteen features.

Discussion

Across the multiclass and subsequent binary classification tasks, both the GBDT and MLP models were able to consistently classify songs by cultural market with moderately high accuracy, and the GBDT was often marginally better than the MLP for this purpose. This suggests that the patterns of difference between cultural markets were robust enough to be detected by two different algorithms. A comparison of accuracy scores suggested that the difference between Chinese and Japanese music afforded higher accuracy to the models, than Japanese and English differences. While this could be for several reasons, we speculate a possibility in that music preferences between Japanese and Chinese cultures differed greater that Japanese and English differences. Such a reasoning would support growing calls for decentralisation and internationalisation of psychological research (Heine & Ruby, 2010; Henrich, Heine & Norenzayan, 2010; Cheung, 2012), in showing that Japanese and Chinese speaking cultures, sometimes thought to be homogenous in cross-cultural research, may actually be more different that previously assumed.

A visual inspection of the PDPs and feature importance scores provide some indicator of where these differences lie. Apart from instrumentalness, all identified feature importance proved to be different across (geographical) cultures in similar directions in the follow-up study. This implies that cultural music preferences are reflected both in the music produced by a culture for their respective industry or market, as well as the overall music preferences by geographically bound members of that culture. While this paper does not empirically explore the underlying cultural mechanisms that may account for these differences, this is nevertheless a starting point for future research to continue from, and we speculate on some interpretations for these results. Western music similarly differed from Chinese and Japanese music through higher speechiness. One explanation could be prosodic bias, in that normal spoken Mandarin Chinese inherently contains more pitch movements than English (Hirst, 2013), and consequently, what may be perceived as ‘speech-like’ by Chinese listeners may not correspond to high speechiness scores. However, this can also be explained through previous research on emotion-arousal preferences in the Western and East-Asian contexts. The high speechiness score in English music could indicate larger preferences for hip-hop and rap music. Rap-music has seen a dramatic increase in popularity in Western markets from the 1980s (Mauch et al., 2015), and has been shown to express and embody high-arousal emotions like anger (e.g., Hakvoort, 2015), and its relative popularity in Anglo-American cultures could be representative on cultural preferences towards these high arousal emotions described earlier (Tsai, 2007), compared to Japanese and Chinese cultures.

One feature that differentiated Chinese from English and Japanese music was high acousticness. This points to lower use of electronic instruments in the production process, and may suggest a preference for more organic, natural sounds in Chinese music. Energy appeared to be more important in Japanese than English or Chinese music. We posit that energy preferences in Japan (defined by Spotify as a combination of loudness, complexity, timbre, dynamic range, and noise) could be due to remnants of traditional music aesthetics, that overlap considerably with energy definitions (e.g., beauty in noise/simplicity in complexity: sawari, wabi-sabi, see Deva, 1999; Anderson, 2014; Okuno, 2015). On the surface, this could be similarly concluded from increased loudness features in Japanese over Chinese music (from the follow-up study), but the U-shaped relationship between energy and Japanese/Chinese music classification seen in Fig. 2 suggests a deeper nuance that requires further research.

Finally, we consider the strengths and limitations our exploratory approach. Comparing music features offer a greater insight into behavioural and consumption patterns of music preference across cultural spheres. In doing so, we uncover systematic differences between groups that, while being consistent with previous literature, also offer new insight into how cultures differ, that future research can build from in understanding societies. Unfortunately, we were unable to eliminate certain sample biases from our dataset: we assumed our Chinese data to be representative of Chinese music in general, but Spotify is not (as of 2021) active in China despite the inclusion of several mainland Chinese artists in the database. Instead, our findings represented Chinese-speaking listeners in Taiwan and Hong Kong, along with possibly Malaysia and Singapore, who may have differing values and preferences from mainland Chinese listeners, particularly given differences in demographics of users and variation in dialect. Moreover, using the Spotify API limited our selection of music features to those available in the API. Future studies could examine music features through publicly available software (e.g., MIRtoolbox; Lartillot, Toiviainen & Eerola, 2008) that have both greater amounts of features and more transparent documentation.

On the other hand, our strengths include our comparisons of features, as opposed to genre, that allowed for validity in comparing cultures because of universality in the perceptual properties of music (Savage et al., 2015). This enabled us to conclude that any differences in music features would be due to preference for those features. By contrast, comparing preferences by genre differences across cultures could have introduced confounds to the investigation, as genre is not homogenous across cultures (see Bennett, 1999).

Conclusions

In sum, we demonstrated the variability of music preferences across Chinese (Taiwan), Japanese, and Western (American/English) cultural markets, and identified the features that best account for these differences. In particular, Chinese music was marked by high acousticness, Anglo-American Western music was marked by high speechiness, and Japanese music appeared to be marked by high energy. While we speculated on some reasons why this would be so, future research is needed to validate these theories to develop a holistic understanding of popular music preferences in Chinese and Japanese cultures. As music is an integral part of human society and culture, understanding the mechanisms by which we prefer different types of music may also shed light on the aspects of human society and experience that correspond to these differences.

Our paper also demonstrates the potential uses of machine learning and other computer science methods in cross-cultural research. Given the advent of digital and online media, these repositories of cultural products may hold valuable insight into the diversity of humanity. Computer science as a field has utilised these kinds of data to great effect, be it in developing recommendation systems, or in predicting consumer behaviour, and we hope to demonstrate that these same data and methods can also contribute towards research on society and culture. While a common argument has been that machine learning emphasises prediction, whereas social scientific research prefers interpretation, we show that the two goals are not mutually exclusive. As demonstrated through the use of model interpretation techniques like RFIs and PDPs, supplementing commonly-used prediction focused models in computer science with explanation and model interpretation techniques enables big data and machine learning to offer an efficient and viable means for the empirical analysis of sociocultural phenomenon. At the same time, these methods are more objective, and hold less bias than commonly used methods like self-reports. Particularly for cross-cultural research, future directions could also apply this methodology to identify new avenues of cultural differences in other mediums, as an additional analysis tool. Additionally, computer scientists and engineers could also benefit from this knowledge, as such analyses of sociocultural phenomenon could also aid with the fine-tuning of weights in more opaque deep learning models, such as in recommendation systems used by streaming companies when targeting users from different cultures.

Additional Information and Declarations

Competing Interests

Author Contributions

Data Availability

The authors declare that they have no competing interests.

Kongmeng Liew conceived and designed the experiments, performed the experiments, analyzed the data, performed the computation work, prepared figures and/or tables, authored or reviewed drafts of the paper, and approved the final draft.

Yukiko Uchida conceived and designed the experiments, authored or reviewed drafts of the paper, and approved the final draft.

Igor de Almeida conceived and designed the experiments, analyzed the data, prepared figures and/or tables, authored or reviewed drafts of the paper, and approved the final draft.

The following information was supplied regarding data availability:

Data and analysis scripts are available at OSF: Liew, Kongmeng. 2021. “Exploring Cultural Differences in Music Preferences for Chinese, Japanese and English Songs through Spotify Features.” OSF. July 7. osf.io/d3cky.

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
