# Peer review of "Cultural differences in music features across Taiwanese, Japanese and American markets"

_PeerJ Computer Science, doi:10.7717/peerj-cs.642_

## Round 0.1 · original submission · Major Revisions

Please take a look at the comments from reviewers.

Reviewer 1 ·

Basic reporting

The title is too ambiguous and grammatically wrong.
The authors have to check term usages on the overall manuscript, e.g., boom in music streaming.

Motivation is not reasonable: "Preferences for music can be represented through music features."
User preferences are combinations of item features and users' tastes.
Please, give some better explanations without logical leaps.

I cannot find the originality of this study.
The authors merely classified musics into market regions by using the conventional ML-based classifiers.
Although the authors said that contributions of musical features to the classification accuracy can show cultural differences between the market regions, they did not provide adequate discussions for this point (just a feature has higher contribution to this market region than the other regions).

There have been numerous studies for classifying musics according to their physical features.
Therefore, the following sentence cannot be the contribution of this study.
"We demonstrate that machine learning can reveal both the magnitude of differences
38 in music preference across Taiwanese, Japanese, and American markets, and where these preferences are different."
Average readers have already known that we can do these things with the conventional ML models.

Experimental design

The experimental subjects and procedures are adequate.
Also, they have been described well.

Validity of the findings

The findings of this paper should be cultural differences between music markets.
However, the authors have concentrated on the effectiveness of the machine learning on classifying musics.
According to this point, the abstract and introduction should be modified.

Also, to reveal the cultural differences, contributions of musical features to the classification accuracy are not enough.
Please, add more in-depth analysis to ensure the originality of this study.

Reviewer 2 ·

Basic reporting

no comment

Experimental design

no comment

Validity of the findings

no comment

Additional comments

The paper is, generally, well-described and clear.
However, Figure 2 and 3 need to be redrawn since they are not clear.

Reviewer 3 ·

Basic reporting

Overall, this research paper is well structured. Although it is a minor thing, authors are suggested to prove read the article to the professional reader to improve the quality of the word choice and the paper's story flow.

Experimental design

The authors define the experiment steps in a good way. However, several things still need to be clarified:
1. The sufficiency of the sample size.
2. The justification of the features that the authors used for classification.
3. The training-testing split ratio whether it is optimum.
4. The tuned cross-validation parameter if it is enough.

Validity of the findings

1. The authors are strongly encouraged to use a statistical measure to compare the model's quality rather than just comparing the accuracy and AUC score. For instance, using Delong's method to compare the model performance.
2. It will be better for the authors to explicitly restate this study's novelty, on how this study differs from other studies and its contribution, in the conclusion part. It could link the finding and the research goal stated initially and make the importance of this study more sound.

---

## Round 0.2 · Major Revisions

Please revise your manuscript based on the review comments. Importantly, authors need to consider how to emphasize the relevance of the work to the topical coverage of this journal. The final decision will be made on the revision.

Reviewer 1 ·

Basic reporting

no comment

Experimental design

no comment

Validity of the findings

no comment

Additional comments

The authors have significantly revised their manuscript according to review comments. The authors have changed their main contribution from classifying popular music according to cultural backgrounds to analyzing popular music.

However, I still do not agree that the analysis methods used in this study have novelty.
Also, the authors have not shown applications or potential applications of the analysis methods.

Therefore, the current version of the manuscript is not suitable for journals in the computer science area.
I want to recommend the authors search for journals dealing with data analysis results, not analysis methodologies.

Reviewer 2 ·

Basic reporting

no comment

Experimental design

no comment

Validity of the findings

no comment

---

## Round 0.3 · accepted · Accept

The paper has been well revised. I think the paper can be accepted.